Integrating precision medicine in the study and clinical treatment of a severely mentally ill person

O’Rawe Jason A. 1 2
Fang Han 1 2
Rynearson Shawn 3
Robison Reid 4
Kiruluta Edward S. 5
Higgins Gerald 6
Eilbeck Karen 3
Reese Martin G. 5
Lyon Gholson J. 1 2 4 GholsonJLyon@gmail.com
1 Stanley Institute for Cognitive Genomics, Cold Spring Harbor Laboratory , NY , USA
2 Stony Brook University , Stony Brook, NY , USA
3 Department of Biomedical Informatics, University of Utah , Salt Lake City, UT , USA
4 Utah Foundation for Biomedical Research , Salt Lake City, UT , USA
5 Omicia Inc. , Emeryville, CA , USA
6 AssureRx Health, Inc. , Mason, OH , USA
Appelbaum Paul
Electronic publication date: 2013 Oct 3
Publication date: 2013
Volume: 1
Electronic Location ID: e177
Received 2013 Jun 12; Accepted 2013 Sep 16
Copyright: © 2013 O’Rawe et al.
Copyright year: 2013
Copyright holder: O’Rawe et al.
License: This is an open access article distributed under the terms of the Creative Commons Attribution License, which permits unrestricted use, distribution, and reproduction in any medium, provided the original author and source are credited.
License URL: https://creativecommons.org/licenses/by/3.0/

Keywords: Genomics, Deep brain stimulation, Whole genome sequencing, Ethics, Neurosurgery, Obsessive compulsive disorder

Funding: NIH SBIR R44HG006579 Stanley Institute for Cognitive Genomics at Cold Spring Harbor Laboratory ESK and MGR are supported by NIH SBIR grant R44HG006579. GJL is supported by funds from the Stanley Institute for Cognitive Genomics at Cold Spring Harbor Laboratory. The funders had no role in study design, data collection and analysis, decision to publish, or preparation of the manuscript.

==============================
Background. In recent years, there has been an explosion in the number of technical and medical diagnostic platforms being developed. This has greatly improved our ability to more accurately, and more comprehensively, explore and characterize human biological systems on the individual level. Large quantities of biomedical data are now being generated and archived in many separate research and clinical activities, but there exists a paucity of studies that integrate the areas of clinical neuropsychiatry, personal genomics and brain-machine interfaces.

Methods. A single person with severe mental illness was implanted with the Medtronic Reclaim® Deep Brain Stimulation (DBS) Therapy device for Obsessive Compulsive Disorder (OCD), targeting his nucleus accumbens/anterior limb of the internal capsule. Programming of the device and psychiatric assessments occurred in an outpatient setting for over two years. His genome was sequenced and variants were detected in the Illumina Whole Genome Sequencing Clinical Laboratory Improvement Amendments (CLIA)-certified laboratory.

Results. We report here the detailed phenotypic characterization, clinical-grade whole genome sequencing (WGS), and two-year outcome of a man with severe OCD treated with DBS. Since implantation, this man has reported steady improvement, highlighted by a steady decline in his Yale-Brown Obsessive Compulsive Scale (YBOCS) score from ∼38 to a score of ∼25. A rechargeable Activa RC neurostimulator battery has been of major benefit in terms of facilitating a degree of stability and control over the stimulation. His psychiatric symptoms reliably worsen within hours of the battery becoming depleted, thus providing confirmatory evidence for the efficacy of DBS for OCD in this person. WGS revealed that he is a heterozygote for the p.Val66Met variant in BDNF, encoding a member of the nerve growth factor family, and which has been found to predispose carriers to various psychiatric illnesses. He carries the p.Glu429Ala allele in methylenetetrahydrofolate reductase (MTHFR) and the p.Asp7Asn allele in ChAT, encoding choline O-acetyltransferase, with both alleles having been shown to confer an elevated susceptibility to psychoses. We have found thousands of other variants in his genome, including pharmacogenetic and copy number variants. This information has been archived and offered to this person alongside the clinical sequencing data, so that he and others can re-analyze his genome for years to come.

Conclusions. To our knowledge, this is the first study in the clinical neurosciences that integrates detailed neuropsychiatric phenotyping, deep brain stimulation for OCD and clinical-grade WGS with management of genetic results in the medical treatment of one person with severe mental illness. We offer this as an example of precision medicine in neuropsychiatry including brain-implantable devices and genomics-guided preventive health care.

Introduction

Deep brain stimulation (DBS) has emerged as a relatively safe and reversible neurosurgical technique that can be used in the clinical treatment of traditionally treatment resistant psychiatric disorders. DBS enables the adjustable and stable electrical stimulation of targeted brain structures. A recent paper by Höflich et al. (2013) notes variability in treatment outcomes for DBS patients, which is likely due to variable responses to differences in targeted stimulation regions and in post-operative stimulation parameters. Both sources of variation, the authors note, will effect the stimulation of different brain tissue fibers having different anatomical and functional connections. Furthermore, the authors suggest that not every target will be suitable for every person, as there exists a large degree of inter-individual variability of brain region activation during a reward task in healthy volunteers, and suggest that future work could (and should) focus on developing surgical plans based on individual-specific activations, functional connectivity and/or tractography. This work exemplifies the large degree of clinically relevant biological variability that exists in terms of individual clinical characteristics.

Ongoing clinical trials testing the “Effectiveness of Deep Brain Stimulation for Treating People With Treatment Resistant Obsessive-Compulsive Disorder” (Greenberg, 2013) detail the below exclusion criteria: • current or past psychotic disorder,

• a clinical history of bipolar mood disorder, and/or

• an inability to control suicide attempts, imminent risk of suicide in the investigator’s judgment, or a history of serious suicidal behavior, which is defined using the Columbia-Suicide Severity Rating Scale (C-SSRS) as either: one or more actual suicide attempts in the 3 years before study entry with the lethality rated at 3 or higher, or one or more interrupted suicide attempts with a potential lethality judged to result in serious injury or death.

These study criteria exclude the most severe cases of OCD, as many people with severe OCD also have severe depression, usually with passive (and sometimes active) suicidal ideation (Torres et al., 2011; Alonso et al., 2010; Balci & Sevincok, 2010). Obsessions and compulsions can be quite severe, with very poor insight, sometimes to a delusional or psychotic degree, and there can also be co-occurring psychoses in any individual. Each person is to some degree unique in his or her psychiatric presentation, and a tailored evaluation schema could prove more effective in clinical treatment. Due in part to these above hurdles, there are few detailed descriptions of the efficacy of DBS for OCD, with the number of published case studies on the efficacy of DBS for OCD covering upwards of ∼100 people (Roh et al., 2012; Goodman & Alterman, 2012; Blomstedt et al., 2012; Burdick & Foote, 2011; Mian et al., 2010; Haynes & Mallet, 2010; Goodman et al., 2010; Denys et al., 2010; Komotar, Hanft & Connolly, 2009; Jimenez-Ponce et al., 2009; Denys & Mantione, 2009; Burdick, Goodman & Foote, 2009; Shah et al., 2008; Figee et al., 2013; Lipsman et al., 2012; Lipsman, Neimat & Lozano, 2007).

An explosive growth in exome and whole genome sequencing (WGS) (Lyon & Wang, 2012) has occurred in parallel to the emergence of DBS for OCD, led in part by dramatic cost reductions. This in turn has given medical practitioners an efficient and comprehensive means to medically assess coding and non-coding regions of the genome, leading to much promise in terms of assessing and treating human disease. In our own efforts to push forward the field of precision medicine, we report here one effort to integrate the areas of clinical neuropsychiatry, brain machine interfaces and personal genomics in the individualized care of one person. We evaluate and treat an individual with DBS for treatment refractory OCD, gauge the feasibility and usefulness of the medical integration of genetic data stemming from whole genome sequencing, and search for rare variants that might alter the course of medical care for this person. As mentioned above, there have been relatively few reports on studies detailing the effective application of DBS for OCD; we report here one such study.

Methods

Ethics compliance

Research was carried out in compliance with the Helsinki Declaration. The corresponding author (GJL) conducted all clinical evaluations and he is an adult psychiatry and child/adolescent psychiatry diplomate of the American Board of Psychiatry and Neurology. GJL obtained IRB approval #00038522 at the University of Utah in 2009–2010 to evaluate candidates for surgical implantation of the Medtronic Reclaim® DBS Therapy for OCD, approved under a Humanitarian Device Exemption (HDE) for people with chronic, severe, treatment-resistant OCD (Medtronic, 2013). The interdisciplinary treatment team consisted of one psychiatrist (GJL), one neurologist and one neurosurgeon. Implantation ultimately occurred on a clinical basis at another site. Written consent was obtained for phenotyping and whole genome sequencing through Protocol #100 at the Utah Foundation for Biomedical Research, approved by the Independent Investigational Review Board, Inc. Informed and written consent was also obtained using the Illumina Clinical Genome Sequencing test consent form, which is a clinical test ordered by the treating physician, GJL.

Evaluation and recruitment for DBS for treatment-refractory OCD

GJL received training regarding DBS for OCD at a meeting hosted by Medtronic in Minneapolis, Minnesota, in September 2009. The same author attended a Tourette Syndrome Association meeting on DBS for Tourette Syndrome, Miami, Florida, in December 2009. Approximately ten candidates were evaluated over a one-year period in 2010. The individual discussed herein received deep brain stimulation surgery at another site, and then returned for follow-up with GJL. Another psychiatrist, author RR, provided ongoing consultation throughout the course of this study. Although other candidates have since returned for follow-up (with GJL), no others have been surgically treated.

CLIA WGS and the management of results from sequencing data

CLIA WGS using the Illumina Individual Genome Sequencing test

Whole genome sequencing was ordered on this individual as part of our ongoing effort to implement precision medicine in the diagnosis, treatment, and preventive care for individuals. His genome was sequenced in the Illumina Clinical Services Laboratory (CLIA-certified, CAP-accredited) as part of the TruSight Individual Genome Sequencing (IGS) test, a whole-genome sequencing service using Illumina’s short-read sequencing technology (Individual Genome Sequencing (IGS) Test, 2013) (Fig. 2). Although clinical genome sequencing was ordered by GJL on a clinical basis (thus not requiring IRB approval), the clinical phenotyping and collection of blood and saliva for other research purposes was approved by the Institutional Review Board (iIRB) (Plantation, Florida) as part of a study protocol at the Utah Foundation for Biomedical Research (UFBR). Consistent with laboratory-developed tests, WGS has not been cleared or approved by the US Food and Drug Administration (Lyon & Segal, 2013). The entire procedure included barcoded sample tracking of the blood collected by GJL from this person, followed by DNA isolation and sequencing in the Illumina CLIA lab. Data statistics are summarized in Fig. S1.

WGS data analysis and variant prioritization

For the bioinformatics analyses, Illumina utilized the internal assembler and variant caller CASAVA (short for Consensus Assessment of Sequence And VAriation). Reads were mapped to the Genome Reference Consortium assembly GRCh37. Data for sequenced and assembled genomes was provided on one hard drive, formatted with the NTFS file system and encrypted using the open source cross platform TrueCrypt software (www.truecrypt.org) and the Advanced Encryption Standard (AES) algorithm (Federal Information Processing Standards Publication 197).

Genotyping array data was generated in parallel of the CLIA whole genome sequencing, using the Illumina HumanOmni2.5-8 bead chip. The encrypted hard drive contains several files, including a “genotyping folder” within which there is a genotyping report in a text-based tab-delimited format (see File S1). See File S11 for more details on the genotyping array data.

Insertions, deletions and structural alterations are not validated variant types in the Illumina Clinical Services Laboratory. Insertions and deletions provided in the gVCF file are for investigative or research purposes only. A medical report and the raw genomic data were provided back to the ordering physician (GJL) on an encrypted hard drive as part of the Illumina Understand your Genome Symposium, held in October 2012, which included the clinical evaluation of 344 genes (see Files S2 and S3) (Friend et al., 2013).

To perform more comprehensive downstream analyses using a greater portion of the genomic data, all of the variants that were detected by the Illumina CLIA WGS pipeline were imported and analyzed within the Omicia Opal web-based clinical genome interpretation platform (Fig. 2, Fig. S5), version 1.5.0 (Omicia, 2013). The Omicia system annotates variants and allows for the identification and prioritization of potentially deleterious alleles. Omicia Scores, which are computationally derived estimates of deleteriousness, were calculated by using a decision-tree based algorithm, which takes as input the Polyphen, SIFT, MutationTaster and PhyloP score(s), and derives an integrative score between 0 and 1. Receiver operating characteristic (ROC) curves are plotted for that score based on annotations from HGMD. For further details on the method and the program see the File S11 and www.omicia.com. The AssureRx Health, Inc. annotation and analysis pipeline was used to further annotate variants (see File S11 for more detailed methods).

We also applied a recently published method, ERDS (Estimation by Read Depth with SNVs) version 1.06.04 (Zhu et al., 2012), in combination with genotyping array data, to generate a set of CNV calls. ERDS starts from read depth information inferred from BAM files, but also integrates other information including paired end mapping and soft-clip signature, to call CNVs sensitively and accurately. We collected deletions and duplications that were > 200 kb in length, with confidence scores of > 300. CNVs that were detected by the ERDS method were visually inspected by importing and visualizing the read alignment data in the Golden Helix Genome Browser, version 1.1.1. CNVs were also independently called from Illumina HumanOmni2.5-8v1 genotyping array data. Array intensities were imported and analyzed within the Illumina GenomeStudio software suite, version 1.9.4. LogR values were exported from GenomeStudio and imported into Golden Helix SVS, version 7.7.5. A Copy Number Analysis Method (CNAM) optimal segmentation algorithm was used to generate a list of putative CNVs, which was then restricted to include only CNVs that were > 200 kb in length with average segment LogR values of > 0.15 and < −0.15 for duplications and deletions, respectively. LogR and covariate values were plotted and visually inspected at all genomic locations where the CNAM method detected a CNV. CNVs that were simultaneously detected by both methods (ERDS and CNAM) were considered to be highly confident CNVs. Highly confident CNVs were, again, visually inspected within Golden Helix Genome Browser to further eliminate any artefactual CNV calls.

Managing sequencing results

There are multiple steps involved in the management of clinical test results, beginning with bar-coded tracking of orders and the return of results to the clinician’s office from the outside CLIA-certified testing facility. The results are transferred to the clinician, who reviews, signs, and interprets the results and incorporates them into the medical health record. The patient is notified, and needed follow-up is arranged.

In an ongoing effort to develop ways to incorporate genomic data into clinical EHR, we also collaborated with the Sequence Ontology Group to convert the data into the GVFclin format (see File S12). The Genome variant format (GVF), which uses Sequence Ontology to describe genome variation (Reese et al., 2010), has been extended for use in clinical applications. This extended file format, called GVFClin (Eilbeck, 2013), adds the necessary attributes to support Health Level 7 compatible data for clinical variants. The GVF format represents genome annotations for clinical applications using existing EHR standards as defined by the international standards consortium: Health Level 7. Thus, GVFclin can describe the information that defines genetic tests, allowing seamless incorporation of genomic data into pre-existing EHR systems.

Results

Pertinent clinical symptoms and treatment

A 37-year old man and US veteran (here named with pseudonymous initials MA) was evaluated by GJL in 2010 for severe, treatment-refractory obsessive compulsive disorder (OCD), which is an illness that can be quite debilitating (Murphy, Zine & Jenike, 2009). MA had a lifelong history of severe obsessions and compulsions, including contamination fears, scrupulosity, and the fear of harming others, with much milder symptoms in childhood that got much worse in his early 20’s. His Yale-Brown Obsessive Compulsive Scale (YBOCS) (Goodman et al., 1989a; Goodman et al., 1989b) ranged from 32 to 40, indicating extremely severe OCD. Perhaps the worst period of OCD included a 5-day, near continuous, period of tapping on his computer keyboard as a compulsion to prevent harm from occurring to his family members. MA had suffered throughout his life from significant periods of depression with suicidal ideation, and he had attempted suicide at least three times. His prior psychiatric history also includes episodes of paranoia relating to anxieties from his OCD, and he continues to be treated with biweekly injections of risperidone. His Global Assessment of Functioning (GAF) typically ranged from 5 to 15 on a 100 point scale.

His treatment history included over 15 years of multiple medication trials, including clomipramine and multiple SSRIs at high doses, including fluoxetine at 80 mg by mouth daily, along with several attempts with outpatient exposure and ritual prevention (ERP) therapy (Gillihan et al., 2012). MA inquired and was evaluated by GJL at the University of Utah and then at two other centers independently offering deep brain stimulation for OCD. One of these centers required (as a condition for eligibility for an ongoing clinical trial) a two-week inpatient hospitalization with intensive ERP, which subsequently occurred and was documented as improving his YBOCS score to 24 at discharge. He maintains that he actually experienced no improvement during that hospitalization, but rather told the therapists what they wanted to hear, as they were “trying so hard”. See the File S11 for other clinical details.

The teams at the University of Utah and two other centers declined to perform surgery due to his prior history of severe depression, suicide attempts and possible psychoses with paranoia. Through substantial persistence of MA and his family members, a psychiatrist and neurosurgeon at a fourth center decided that he was an appropriate candidate for surgical implantation of the Medtronic Reclaim® DBS Therapy device for OCD, approved under a Humanitarian Device Exemption (HDE) for people with chronic, severe, treatment-resistant OCD (Medtronic, 2013), and he was implanted in January of 2011 (Fig. 1). The device targets the nucleus accumbens/anterior limb of the internal capsule (ALIC). A detailed account of the surgical procedure can be found in the File S11.

Figure 1 Sagittal and transverse computed tomography (CT) images of the brain and skull of MA.

We show here sagittal and transverse sections taken from CT scans. Imaging was performed before (A) and after (B) MA received deep brain stimulation surgery for his treatment refractory OCD. Two deep brain stimulator probes can be seen to be in place from a bifrontal approach (B), with tips of the probes located in the region of the hypothalamus. Leads traverse through the left scalp soft tissues. Streak artifact from the leads somewhat obscures visualization of the adjacent bifrontal and left parietal parenchyma. We did not observe any intracranial hemorrhage, mass effect or midline shift or extra-axial fluid collection. Brain parenchyma was normal in volume and contour.

Figure 2 Implementation of the analytic-interpretive split model for the clinical incorporation of a whole genome.

We have implemented the analytic-interpretive split model here with MA, with WGS being performed in a CLIA certified and CAP accredited lab at Illumina as part of the Individual Genome Sequencing test developed by them. The WGS acts as a discrete deliverable clinical unit from which multiple downstream interpretive analyses were performed. We used the ERDS CNV caller, the Golden Helix SVS CNAM for CNV calling, and the Omicial Opal and the AssureRx Health Inc. pipelines for variant annotation and clinical interpretation of genomic variants. By archiving and offering to him the encrypted hard drive containing his “raw” sequencing data, any number of people, including the individual and/or his/her health care providers can analyze his genome for years to come. Abbreviations: CLIA, Clinical Laboratory Improvement Amendments; CAP, College of American Pathologists; CASAVA, Consensus Assessment of Sequence and Variation; ERDS, Estimation by Read Depth with SNVs; CNAM, Copy Number Analysis Method; WGS, Whole Genome Sequencing.

Clinical results for DBS for treatment-refractory OCD

After healing for one month, the implanted device (equipped with the Kinetra Model 7428 Neurostimulator) was activated on February 14, 2011, with extensive programming by an outpatient psychiatrist, with bilateral stimulation of the ALIC. Final settings were case positive, contact 1 negative on the left side at 2.0 V, frequency 130 Hz, and pulse width 210 µs, and case positive, contact 5 negative on the right side with identical settings.

Over the next few months, his voltage was increased monthly in increments of 0.2–0.5 V by an outpatient psychiatrist. He returned to one of the author’s (GJL) for psychiatric treatment in July 2011, at which time his voltage was set at 4.5 V bilaterally. His depression had immediately improved after the surgery, along with many of his most irrational obsessions, but his YBOCS score still remained in the 35–38 range. From July 2011–December 2011, his voltage was increased bilaterally on a monthly basis in increments of 0.2 V, with steady improvement with his OCD until his battery started to lose charge by December 2011. This caused him considerable anxiety, prompting him to turn off his battery in order to “save battery life”, which unfortunately led to a complete relapse to his baseline state in a 24 h period, which was reversed when he turned the battery back on. The battery was surgically replaced with a rechargeable Activa RC neurostimulator battery in January 2012, and the voltage has been increased monthly in 0.1–0.2 V increments until the present time (May 2013).

Figure 3 Yale Brown Obsessive Compulsive Scale (YBOCS) scores were measured for MA over a three year and seven months period of time.

A time series plot (A) shows a steady decline in YBOCS scores over the period of time spanning his DBS surgery (s) and treatment. Incremental adjustments to neurostimulator voltage are plotted over a period of time following DBS surgery. Mean YBOCS scores are plotted for sets of measurements taken before and after Deep Brain Stimulation (DBS) surgery (B). A one-tailed unpaired t test with Welch’s correction results in a p value of 0.0099, demonstrating a significant difference between YBOCS scores measured before and after the time of surgery.

At every visit, MA has reported improvements, with reductions of his obsessions and compulsions, marked by an overall decline in his YBOCS score (Fig. 3). MA has started to participate in many activities that he had never previously been able to engage in. This includes: exercising (losing 50 pounds in two years) and volunteering at the church and other organizations, but not yet being able to work in any paid capacity. MA also started dating and recently got married, highlighting his improvement in daily functioning, with a GAF score ranging now from 40–50. New issues that MA reports are consistent tenesmus, occasional diarrhea (which he can now tolerate despite prior contamination obsessions) and improved vision (going from 20/135 to 20/40 vision, as documented by his optometrist), with him no longer needing to wear glasses. It is unknown whether the DBS implant has contributed to any of these issues. Attempts to add fluoxetine at 80 mg by mouth daily for two months to augment any efficacy from the DBS and ERP were unsuccessful, mainly due to no discernible benefit and prominent sexual side effects. MA still receives an injection of 37.5 mg risperidone every two weeks for his past history of psychoses; otherwise, he no longer takes any other medications. There has not been any exacerbation of psychoses in this individual during the two years of treatment with DBS.

CLIA certified WGS results

Illumina WGS clinical evaluations

The Illumina WGS clinical evaluation included manual annotation of 344 genes (see Fig. S2, Files S2 and S3), which led to the following conclusion:

“No pathogenic or likely pathogenic variants were found in the 344 genes evaluated that are expected to be clinically significant for the patient. The coverage for these 344 genes is at least 99%. Therefore, significant variants could exist that are not detected with this test.”

The clinical evaluation did, however, identify MA as a carrier for a variant (c.734G>A, p.Arg245Gln) in PHYH, which has been associated in the autosomal recessive or compound heterozygote states with Refsum disease, which is an inherited condition that can lead to vision loss, anosmia, and a variety of other signs and symptoms (Greenberg et al., 2006). In silico prediction programs suggest little impact; however, the variant is rare with a 1000 Genomes frequency of ∼0.18%. In this regard, it is worth noting that MA has always had poor night vision and enlarged pupils, and, as a result of this genetic finding, we met with MA’s treatment team at his Veteran’s Affair’s (V.A.) medical center and learned that he had recently been diagnosed with bilateral cataracts, enlarged pupils, and vision loss. We also learned that MA’s mother and maternal grandfather have a history of enlarged pupils with poor vision, and we are currently following up whether this might be related in any way to this particular variant and Refsum disease.

Disease variant discovery

Further downstream analyses (Fig. 2) identified and prioritized several other potentially clinically relevant variants. Variants that were imported into the Omicia Opal system were filtered to include those that had a high likelihood of being damaging (as defined by an Omicia score >0.7) and those that have supporting Online Mendelian Inheritance in Man (OMIM; an online database of human genetics and genetic disorders) evidence. We chose to filter based on an Omicia Score of > 0.7 as this value derives a slightly more inclusive list of variants which still cannot be dismissed, but for which we have additional corroborating evidence (i.e., Illumina Genome Network (IGN) validation and annotation). These prioritized variants were further annotated and evaluated by the AssureRx Health, Inc. annotation and analysis pipeline. Prioritized variants are shown in File S4 and Fig. S3. A longer list of variants, which were required only to have supporting evidence within the OMIM database, is shown in File S5. We highlight here some of the findings:

MA was found to be a heterozygote for a p.Val66Met change in BDNF, which encodes a protein that is a member of the nerve growth factor (NGF) family. The protein is induced by cortical neurons, and is deemed necessary for the survival of striatal neurons in the brain. In drug naïve patients, BDNF serum levels were found to be significantly decreased in OCD patients when compared to controls (36.90 ± 6.42 ng/ml versus 41.59 ± 7.82 ng/ml; p = 0.043) (Maina et al., 2010), suggesting a link between this protein and OCD. Moreover, a study including 164 proband-parent trios with obsessive-compulsive disorder (Hall et al., 2003) uncovered significant evidence of an association between OCD and all of the BDNF markers that were tested, including the exact variant found here in this person, p.Val66Met. This particular variant has been further studied in a sample of 94 nuclear families (Rosa et al., 2006), which included 94 probands with schizophrenia-spectrum disorders and 282 family members. The results of this study suggest that the p.Val66Met polymorphism may play a role in the phenotype of psychosis. Similar anxiety-related behavioral phenotypes have also been observed among mice and humans having the p.Val66Met variant in BDNF (Soliman et al., 2010). In humans, the amygdala mediates conditioned fear (Davis, 1992), normally inhibited by ‘executive centers’ in medial prefrontal cortex (Moscarello & LeDoux, 2013). Deep brain stimulation of the pathways between medial prefrontal cortex and the amygdala increased the extinction of conditioned fear in a rat model of OCD (Rodriguez-Romaguera, Do Monte & Quirk, 2012). Studies using functional magnetic resonance imaging (fMRI) demonstrate that humans with the p.Val66Met variant exhibit exaggerated activation of the amygdala in response to an emotional stimulus in comparison to controls lacking the variant (Montag et al., 2008; Lau et al., 2010). It is thought that this variant may influence anxiety disorders by interfering with the learning of cues that signal safety rather than threat and may also lessen efficacy of treatments that rely on extinction mechanisms, such as exposure therapy (Soliman et al., 2010). In this regard, it is interesting to note that this person did indeed obtain very little benefit from exposure therapy prior to surgery.

MA heterozygously carries the p.Glu429Ala allele in MTHFR, encoding a protein that catalyzes the conversion of 5,10-methylenetetrahydrofolate to 5-methyltetrahydrofolate, a co-substrate for homocysteine remethylation to methionine, and which has been shown to confer an elevated susceptibility to psychoses. Variants in MTHFR influence susceptibility to occlusive vascular disease, neural tube defects, colon cancer and acute leukemia. Variants in this gene are associated with methylenetetra-hydrofolate reductase deficiency. In addition, a meta-analysis comparing 1,211 cases of schizophrenia with 1,729 controls found that the MTHFR p.Glu429Ala allele was associated with susceptibility to schizophrenia (Allen et al., 2008) (odds ratio, 1.19; 95% CI, 1.07–1.34; p = 0.002). According to the Venice guidelines for the assessment of cumulative evidence in genetic association studies, the MTHFR association exhibited a strong degree of epidemiologic credibility (Frayling, 2008). Pharmacogenetic studies have found a consistent association between the MTHFR p.Glu429Ala allele and metabolic disorder in adult, adolescent and children taking atypical antipsychotic drugs (Correll et al., 2009; van Winkel et al., 2010).

MA is also heterozygous for the p.Val108Met variant in catechol-O-methyltransferase (COMT), which catalyzes the transfer of a methyl group from S-adenosylmethionine to catecho- lamines, including the neurotransmitters dopamine, epinephrine, and norepinephrine. The minor allele A of this 472G>A variant produces a valine to methionine substitution, resulting in a less thermostable COMT enzyme that exhibits a 3-fold reduction in activity. A substantial body of literature implicates this variant as possibly elevating the risk for various neuropsychiatric disorders in some Caucasian populations but not necessarily in other genetic backgrounds (Collip et al., 2011; Dumontheil et al., 2011; Lajin et al., 2011; Raznahan et al., 2011; Lopez-Garcia et al., 2012; Singh et al., 2012; Ira et al., 2013). There is some evidence that MTHFR × COMT genotype interactions might also be occurring in MA to influence his neuropsychiatric status (Roffman et al., 2008), and the same is true for BDNF × COMT interactions (Alonso et al., 2013).

Pharmacogenetic variants

Pharmacogenetic analyses were performed using the Omicia Opal platform. Pharmacogenetic variants were identified and prioritized by activating the “Drugs and Pharmacology” track within the Opal system and by requiring these variants to have prior evidence within any one of several supporting databases (i.e., OMIM, HGMD, PharmGKB, LSDB and GWAS). Prioritized variants are shown in File S6 and Fig. S4. A longer, more inclusive list is shown in File S7; variants in this file are only required to be detected by the “Drugs and Pharmacology” track in Opal. Variants within Files S6/S7 were further annotated and analyzed by the AssureRx Health, Inc. pipeline (see File S8). Below, we highlight pharmacogenetic variants found to be informative in terms of future medication choices for MA.

MA is heterozygous for a c.19G>A p.Asp7Asn allele in ChAT, encoding choline O-acetyltransferase, which synthesizes the neuro-transmitter acetylcholine (Fig. S5). This particular variant (rs1880676) is significantly associated with both risk for schizophrenia in Caucasians (P = 0.002), olanzapine response (P = 0.04) and for other psychopathology (P = 0.03) (Mancama et al., 2007). Allele A is associated with increased response to olanzapine in people with schizophrenia as compared to allele G. This association was nominally significant (p = 0.04) in the Spanish cohort (Mancama et al., 2007).

MA is homozygous for a p.Ile359Leu change in CYP2C9, and this variant has been linked to a reduction in the enzymatic activity of CYP2C9 (Lundblad et al., 2005). CYP2C9 encodes a member of the cytochrome P450 superfamily of enzymes. Cytochrome P450 proteins are mono-oxygenases, which catalyze many reactions associated with drug metabolism as well as reactions associated with the synthesis of cholesterol, steroids and other lipids (Sim & Ingelman-Sundberg, 2013). CYP2C9 localizes to the endoplasmic reticulum and its expression is induced by rifampin. CYP2C9 is known to metabolize xenobiotics, including phenytoin, tolbutamide, ibuprofen as well as S-warfarin. Studies identifying individuals who are poor metabolizers of phenytoin and tolbutamide suggest associations between metabolism and polymorphisms found within this gene. CYP2C9 is located within a cluster of cytochrome P450 genes on chromosome 10 (Nelson et al., 2004). Fluoxetine is commonly used in the treatment of OCD; it has been shown to be as effective as clomipramine and causes less side effects (Pigott & Seay, 1999; Pigott et al., 1990). CYP2C9 acts to convert fluoxetine to R-norfluoxetine (Ring et al., 2001), and so MA may not be able to adequately biotransform fluoxetine (Zhou, Liu & Chowbay, 2009). However, CYP2C9 does not play a rate-limiting role for other SSRIs or clomipramine. In his own treatment experience, MA had no response to an 80 mg daily dose of fluoxetine, although he did experience sexual side effects at that dosage.

The protein encoded by DPYD is a pyrimidine catabolic enzyme and it acts as the initial and rate-limiting factor in uracil and thymidine catabolism pathways. MA was found to be a carrier of two variants in this gene, p.Ile543Val and p.Arg29Cys, for which he is a heterozygote and homozygote, respectively. Variants within DPYD result in dihydropyrimidine dehydrogenase deficiency, an error in pyrimidine metabolism associated with thymine-uraciluria and an increased risk of toxicity in cancer patients receiving 5-fluorouracil chemotherapy. Two transcript variants encoding different isoforms have been described for DPYD (Johnson et al., 1997; Shestopal, Johnson & Diasio, 2000).

MA is heterozygous both for a c.590G>A p.Arg197Gln allele (rs1799930) and a c.803G>A p.Arg268Lys allele (rs1208) in NAT2, encoding an enzyme that functions to both activate and deactivate arylamine and hydrazine drugs and carcinogens. Genotype AG for rs1799930 is associated with increased risk of toxic liver disease in people with tuberculosis when treated with ethambutol, isoniazid, pyrazinamide and rifampin as compared to genotype GG. Allele G for rs1208 is not associated with risk of hypersensitivity when treated with sulfamethoxazole and trimethoprim in people with infection (Sacco et al., 2012; Bozok Cetintas et al., 2008).

Copy number variants

ERDS identified 60 putative CNVs, all of which were visually inspected within the Golden Helix Genome Browser. Many of the CNVs detected by the ERDS method were found to be located within chromosomal boundary regions and were determined to be false positives due to highly variable read depth in these regions. The CNAM method detected 35 putative CNVs, which were visually inspected by plotting the LogR and covariate values in Golden Helix SVS. Only six CNVs were simultaneously detected by both the ERDS and CNAM methods, and were visually inspected as further confirmation to be included among the set of highly confident CNVs. High-confidence CNVs are shown in File S9. To our knowledge, these CNVs have not been previously associated in any way with MA’s disease phenotype, but we are archiving these results for future analysis as knowledge of CNVs and disease associations expands.

Return of results

A board-certified genetic counselor was consulted by GJL prior to returning results, and all therapy and counseling was provided by GJL. Although we believe in archiving and managing all genetic results and not just a small subset of genes, we did analyze the 57 genes that are currently recommended for “return of results” by the American College of Medical Genetics (Green et al., 2013). These results are shown in File S10, and one of us (GJL) met with MA to go over the results with him, along with adding some of the findings into his paper-based medical record. Lastly, we did contact the physicians and other officials at the US Veterans Affairs office to offer to incorporate these data into the electronic medical record for MA at the VA, but we were informed that the VistA health information system (HIS) (Conn, 2011; Protti & Groen, 2008; Kuzmak & Dayhoff, 1998; Brown et al., 2003) does not currently have the capability to incorporate any genomic variant data.

Discussion

DBS for treatment-refractory OCD

Deep brain stimulation for MA’s treatment refractory OCD has provided a quantifiable and significant improvement in the management of his symptoms (Fig. 3). MA has regained a quality of life that he had previously not experienced in over 15 years, which is highlighted by his participating in regular exercise, working as a volunteer in his local church, dating, and eventually getting married, all of which illustrate a dramatic improvement in his daily functioning since receiving DBS treatment for his OCD.

One significant aspect of this study is the rechargeable, and hence depletable, nature of the Activa RC neurostimulator battery, which serves to illustrate the efficacy of DBS for OCD for this individual. On one such illustrative occasion, MA forgot to take the recharging device on a four-day weekend trip. Once his battery was depleted, all of his symptoms gradually returned to their full level over a ∼24 h period, including severe OCD, depression and suicidality. Since that episode, MA always takes his recharging device with him on extended trips, but there have been other such instances in which his battery has become depleted for several hours, with the noticeable and intense return of his OCD symptoms and the cessation of his tenesmus. The electrical stimulation is having a demonstrable effect on his OCD, and these data are complementary to other data-sets involving turning DBS devices off for one week at a time (Figee et al., 2013).

There are many ethical and regulatory issues relating to deep brain stimulation that have been discussed elsewhere (Fins, Dorfman & Pancrazio, 2012; Synofzik, Fins & Schlaepfer, 2012; Fins et al., 2011a; Fins et al., 2011b; Fins & Schiff, 2010; Fins, 2010; Erickson-Davis, 2012), and we report here our one positive experience, made possible when the US Food and Drug Administration granted a Humanitarian Device Exemption (HDE) to allow clinicians to use this intervention. The rechargeable nature of the new battery has been reassuring to MA, as he is able to exert control over his battery life, whereas he previously had no control with the original “single-use” battery that must be replaced when the battery depletes (usually at least once annually). We assume that other persons treated with DBS for OCD will likely also start receiving rechargeable batteries. In this regard, it is worth noting that the recent development of an injectable class of cellular-scale optoelectronics paves the way for implanted wireless devices (Kim et al., 2013), and we fully expect that there will be more brain-machine neural interfaces used in humans in the future (Alivisatos et al., 2013a; Alivisatos et al., 2013b; Pais-Vieira et al., 2013; Thomson, Carra & Nicolelis, 2013; Nicolelis, 2012).

Clinical WGS

There are still many challenges in showing how any one mutation can contribute toward a clear phenotype, particularly in the context of genetic background and possible environmental influences (Moreno-De-Luca et al., 2013). Bioinformatics confounders, such as poor data quality (Nielsen et al., 2011), sequence inaccuracy, and variation introduced by different methodological approaches (O’Rawe et al., 2013) can further complicate biological and genetic inferences. Although the variants discussed in the results section of our study have been previously associated with mental disease, we caution that the data presented are not sufficient to implicate any particular mutation as being necessary or sufficient to lead to the described phenotype, particularly given that mental illness results from a complex interaction of any human with their surrounding environment and social support structures. The genetic architecture of most neuropsychiatric illness is still largely undefined and controversial (Klei et al., 2012; Mitchell & Porteous, 2011; Mitchell, 2012; Visscher et al., 2011). We provide our study as a cautionary one: WGS cannot act as a diagnostic and prognostic panacea for neuropsychiatric disorders, but instead could act to elucidate risk factors for psychiatric disease and pharmacogenetic variants that can inform future medication treatments.

During our study, we found that MA carries at least three alleles that have been associated with neuropsychiatric phenotypes, including variants in BDNF, MTHFR, and ChAT (Table 1). And, although we have discovered informative pharmacogenetic variants in this person, these discoveries have not led to the immediate alteration of this person’s medication schema. We have archived these discoveries, as described below, and expect that these variants will be useful over the course of his life-long medical care. We feel that this information is inherently valuable, as one can never predict with certainty what the future might hold, and a more complete medical profile on individual patients will facilitate more informed medical choices.

Table 1 A summary of three clinically relevant alleles found in the sequencing results of MA.

Variations in MTHFR, BDNF, and ChAT were found to be of potential clinical relevance for this person as they are all implicated in contributing to the susceptibility and development of many neuropsychiatric disorders that resemble those present within MA. A brief summary of the characteristics of each variation is shown, including the gene name, genomic coordinates, amino acid change, zygosity, variation type, estimated population frequency and putative clinical significance.

Gene name	Genomic
coordinates	Amino acid
change	Zygosity	Variation
type	Population
frequency	Clinical significance	
MTHFR	chr1: 11854476	Glu > Ala	heterozygous	non-synon	T:77% G:23%	Susceptibility to psychoses, schizophrenia occlusive vascular disease, neural tube defects, colon cancer, acute leukemia, and methylenetetra-hydrofolate reductase deficiency	
BDNF	chr11: 27679916	Val > Met	heterozygous	non-synon	C:77% T:23%	Susceptibility to OCD, psychosis, and diminished response to exposure therapy	
CHAT	chr10: 50824117	Asp > Asn	heterozygous	non-synon	G:85% A:15%	Susceptibility to schizophrenia and other psychopathological disorders.	

Integrating WGS data into the electronic medical health record

In the context of the incomplete, and sometimes proprietary, nature of human gene mutation databases, it is likely that analyses and medical guidance gleaned from these WGS data will differ from institution to institution. It is therefore important that people be given the opportunity, like with many other traditional medical tests, to obtain “second opinions”. For this to be possible, one must accurately describe the contents of short-read sequencing data in terms of the existing electronic medical health standards, so that these data can be incorporated into an electronic medical health record. Accurately describing the contents of next generation sequencing (NGS) results is particularly critical for clinical analysis of genomic data. However, genomics and medicine use different and often incompatible terminologies and standards to describe sequence variants and their functional effects. In our efforts to treat this one person with severe mental illness, we have implemented the GVFclin format for the variants that were discovered during the sequencing of his whole genome (see File S12). We hope to eventually incorporate his genetic data into his electronic health record if and when the VistA health information system (HIS) (Conn, 2011; Protti & Groen, 2008; Kuzmak & Dayhoff, 1998; Brown et al., 2003) is upgraded to allow entry of such data. We did already counsel MA regarding several genetic variants that may be clinically relevant to predisposing him to his psychiatric disorder (Biesecker & Peay, 2013).

Returning genetic results

There is considerable controversy in the field of medical genetics concerning the extent of return of genetic results to people, particularly in the context of “secondary”, “unrelated”, “unanticipated” or “incidental” findings stemming from new high-throughput sequencing techniques (Lyon, 2012c). Some people have concerns regarding the clinical utility of much of the data, and in response have advocated for selectively restricting the returnable medical content. One such set of recommendations has been provided by the American College of Medical Genetics which recently released guidelines in which they recommended the “return of secondary findings” for 57 genes, without detailed guidance for the rest of the genome (Green et al., 2013). These types of recommendations take a more paternalistic approach in returning test results to people, and generally involve a deciding body of people that can range in size from a single medical practitioner to a committee of experts. We believe that anyone should be able to access and manage their own genome data (Yu et al., 2013), just like how anyone should be able to own and manage their medical and radiology test results (Strata Conference, 2013), particularly if the testing is performed with suitably appropriate clinical standards in place, i.e., CLIA in America (Lyon, 2012a; Lyon, 2012b). In this regard, we found by means of WGS that MA carries a variant in PHYH; this revelation ended up improving his care despite not being related in any known or direct way to his psychiatric disorder. As stated in our Results section, MA has been diagnosed with bilateral cataracts and has been counseled in ways to reduce further damage to and deterioration of his vision.

Conclusions

One can learn a substantial amount from detailed study of particular individuals (for just a small sampling, see Sacks, 1995; Sacks, 1998; Luria, 1972; Luria, 1976; Van Horn et al., 2012; Ratiu et al., 2004; Eichenbaum, 2013; Worthey et al., 2011), and we believe that we are entering an era of precision medicine in which we can learn from and collect substantial data on informative individual cases. Incorporating insights from a range of scientific and clinical disciplines into the study and treatment of any one person is therefore beginning to emerge as a tractable, and more holistic, approach, and we document here what we believe to be the first integration of deep brain stimulation and whole genome sequencing for precision medicine in the evaluation, treatment and preventive care for one severely mentally ill individual, MA. We have shown that DBS has been successful in aiding in the care and beneficial clinical outcome of his treatment refractory OCD, and we have also demonstrated that it is indeed feasible, given current technologies, to incorporate health information from WGS into the clinical care of one person with severe mental illness, including with the return of these health information to him directly. On a comparative level, deep brain stimulation has thus far been a more direct and effective intervention for his mental illness than anything discovered from his whole genome sequencing. Despite this, health information stemming from these WGS data was nevertheless immediately useful in the care of this person, as a variant associated with his ophthalmologic phenotype did indeed inform and enrich his care, and we expect that these data will continue to inform his care as our understanding of human biology and the genetic architecture of disease improves. Of course, the genomic data would have been more helpful if obtained much earlier in his medical course as it could have provided guidance on which medications to avoid or to provide in increased doses.

Supplemental Information

Figure S1 Data statistics and SNP characteristics for the Illumina CLIA WGS pipeline

WGS was performed using the Illumina CLIA WGS pipeline. We report the volume of data, the quality of the data as well as whole genome SNP characteristics and more general characteristics of SNVs reported by the Illumina CLIA WGS pipeline, including: the total number of SNVs, the total number of SNVs that are within genes, coding regions, UTRs, splice site regions as well as the number of SNVs that were stop gained, stop lost, non-synonymous, synonymous and mature mRNA.

Click here for additional data file.

File S1 Genotyping was performed as part of the Illumina CLIA WGS pipeline using the HumanOmni2.5-8v1 BeadChip

The genotyping report is included as a tab-delimited text file and includes a header followed by a number of columns that describe the data within, including: the SNP name, GC score, Allele A – Forward, Allele B – Forward, Allele A – Design and Allele B – Design. The allele calls for the genotyping array are listed in forward orientation.

Click here for additional data file.

Figure S2 Illumina CLIA Whole genome sequencing data summarized in the form of a Circos plot

We show here a summary of the genomic coordinates corresponding to the 344 genes that were clinically evaluated by the Illumina CLIA WGS pipeline, the frequency of IGN validated SNVs across the genome (plotted in red) and a summary of highly confident copy number variants (CNVs) that were simultaneously detected by the Estimation by Read Depth with SNVs (ERDS) and Copy Number Analysis Method (CNAM) detection methods (plotted in black). Duplications and deletions are depicted as elevations and declinations, respectively.

Click here for additional data file.

File S2 The Illumina CLIA WGS clinical report, which includes the clinical evaluation of 140 conditions associated with 344 genes

Clinical interpretation was performed using interpretation guidelines and recommendations from the American College of Medical Genetics. A cumulative total of 1247 variants were detected and evaluated for clinical significance with one single variant being determined as “likely pathogenic”, a p.Arg245Gln change in PHYH.

Click here for additional data file.

Figure S3 Screen shot of the Omicia Opal system showing the list of prioritized variants

Click here for additional data file.

File S3 The 344 gene list analyzed by Illumina as part of the Understand your Genome Symposium in 2012

Click here for additional data file.

Figure S4 Screen shot of the Omicia Opal system showing the list of prioritized pharmacogenetic variants

Click here for additional data file.

File S4 Variant prioritization was performed on all variants discovered by the Illumina CLIA WGS pipeline using the Omicia Opal version 1.5.0 platform

Variants were imported into the Omicia Opal cloud based clinical annotation and variant prioritization platform, and subsequently prioritized by requiring each variant to have prior evidence in OMIM and by additionally requiring each variant to be scored as having an Omicia Score of greater than 0.7.

Click here for additional data file.

Figure S5 Screen shot of a Gene Summary from the Omicia Opal system on ChAT, encoding choline O-acetyltransferase, which synthesizes the neurotransmitter acetylcholine

Omicia Opal was used prioritize and identify genetic variations contained within the whole genome sequence of MA that might be of potential clinical relevance to his neuropsychiatric phenotype. In this figure, we highlight the Opal system as being one method by which clinicians can scan genetic data for clinically relevant information in a robust and comprehensive way. We demonstrate the Opal system with one such variant in ChAT, a heterozygous Asp > Asn variation on chromosome 11.

Click here for additional data file.

File S5 Less stringent variant prioritization was performed on all variants discovered by the Illumina CLIA WGS pipeline using the Omicia Opal version 1.5.0 platform

A more inclusive set of variants was derived by performing less stringent prioritization on all genomic variants. Variants called by the Illumina CLIA WGS and bioinformatics pipeline were imported into the Omicia Opal clinical variant annotation and prioritization platform. Variants were then prioritized by only requiring each variant to have supporting evidence in OMIM.

Click here for additional data file.

File S6 Variants discovered by the Illumina CLIA WGS pipeline to have pharmacogenomic significance were evaluated and prioritized using the Omicia Opal version 1.5.0 platform

Pharmacogenomic variant prioritization was performed by importing all variants called by the Illumina CLIA WGS and bioinformatics pipeline into the Omicia Opal cloud based variant prioritization platform. Variants were filtered by activating the “Drugs and Pharmacology” track in Opal, and further filtered to those that also had prior evidence in a variety of supporting databases, including: OMIM, HGMD, PharmGKB, LSDB and GWAS.

Click here for additional data file.

File S7 Variants discovered by the Illumina CLIA WGS pipeline to have pharmacogenomic significance were evaluated using less conservative prioritization requirements

Less stringent pharmacogenomic variant prioritization was performed by first importing variants called by the Illumina CLIA WGS sequencing and bioinformatics pipeline into the Omicia Opal cloud based variant prioritization platform version 1.5.0. Variants were then filtered only by activating the “Drugs and Pharmacology” track in Opal.

Click here for additional data file.

File S8 Expert curation of pharmacogenetic variants identified by Omicia Opal pipeline

All single nucleotide variants (SNPs), copy number variants (CNVs), indels and other variants identified as important and provided by Omicia’s Opal Annotation Pipeline were investigated further using GenomePharm and manual review of the published literature and clinical trial data. These were compared to genes listed in Pharma DMET, Pharma ADME, and the ADME Pharma consortium. Comparisons were made of known gene-variant-drug-disease interactions as identified in GenomePharm and stored in a temporary NoQL database. Judgments were made by GH, in which a given pharmacogenomic variant had to have been replicated >6 times in adequately powered randomized controlled trials using individuals of European American ancestry.

Click here for additional data file.

File S9 A list of 6 high confidence copy number variants (CNVs) that were called by both the ERDS and CNAM CNV detection methods

ERDS (version 1.06.04) derived CNVs were required to be >200 kb in length, with confidence scores of >300. CNAM (Golden Helix SVS version 7.7.5) CNVs were also required to be >200 kb in length with average segment LogR values of > 0.15 and < −0.15 for duplications and deletions, respectively. CNVs detected by both methods were visually inspected to eliminate obvious false positive calls. The 6 CNVs shown here were detected by each method, visually confirmed, and are thus considered high confidence.

Click here for additional data file.

File S10 57 genes recommended by the ACMG as candidates for returning results were analyzed and annotated by the Omicia Opal system

Only two variants, one in CACNA1S and one in MYLK, were interpreted as being of putative interest but not rising to the level of “pathogenicity”.

Click here for additional data file.

File S11 Supplementary Methods and Clinical Descriptions

Click here for additional data file.

File S12 A GVFclin file of the variants that were detected and validated by the Illumina CLIA whole genome sequencing and bioinformatics pipeline

Sequence variants contained within the GVFclin file format are represented in a way that conforms to the existing electronic medical health standards as defined by the international standards consortium: Health Level 7. The GVFclin file format extends the GVF format by including clinical variant attributes, which are HL7 compatible.

Click here for additional data file.

We thank the many doctors and other caregivers who have worked with MA, including Drs. Paul House, Ziad Nahas and Istvan Takacs. GJL thanks Kenyon Fausett (Medtronic) and Lauren Schrock (University of Utah) for helping train him to perform DBS programming. MA and his family have been extraordinarily cooperative throughout the course of treatment. We also thank Tina Hambuch, Erica Ramos, Dawn Barry and others at Illumina for helping to develop the TruSight Individual Genome Sequencing (IGS) test, a whole-genome sequencing service using Illumina’s short-read sequencing technology in the CLIA-certified, CAP-accredited Illumina Clinical Services Laboratory. They provided fee-for-service whole genome sequencing in the CLIA lab at Illumina, along with generating the clinical report on 344 genes. Julianne O’Daniel graciously provided advice regarding genetic counseling, along with helping to interpret findings in the 57 genes that are currently recommended for “return of results” by the American College of Medical Genetics.

Additional Information and Declarations

Competing Interests

Author Contributions

Human Ethics

DNA Deposition

GJL has had informal discussions with representatives from Medtronic, Illumina, and Omicia, Inc., but he has not had any formal consulting role, nor received financial compensation or grants from these or any other for-profit companies performing deep brain stimulation, DNA collection or sequencing. GJL does not hold any patents, and he is unaware of any conflicts of interest on his part. Revenue earned by GJL from providing medical care in Utah is currently donated to the Utah Foundation for Biomedical Research for genetics research. ESK and MGR are co-founders and officers of Omicia, Inc., and GH is an employee of Assure Rx, Inc. All authors read and approved of the content in the manuscript.

Jason A. O’Rawe performed the experiments, analyzed the data, wrote the paper.

Reid Robison provided psychiatric consultation.

Edward S. Kiruluta and Martin G. Reese analyzed the data, contributed reagents/materials/analysis tools.

Gerald Higgins analyzed the data, contributed reagents/materials/analysis tools.

Gholson J. Lyon performed clinical evaluations, conceived and designed the experiments, performed the experiments, analyzed the data and wrote the paper.

The following information was supplied relating to ethical approvals (i.e., approving body and any reference numbers):

Research was carried out in compliance with the Helsinki Declaration. GJL conducted all clinical evaluations and he is an adult psychiatry and child/adolescent psychiatry diplomate of the American Board of Psychiatry and Neurology. GJL obtained IRB approval #00038522 at the University of Utah in 2009–2010 to evaluate candidates for surgical implantation of the Medtronic Reclaim® DBS Therapy for OCD, approved under a Humanitarian Device Exemption (HDE) for people with chronic, severe, treatment-resistant OCD. The interdisciplinary treatment team consisted of one psychiatrist (GJL), one neurologist and one neurosurgeon. Implantation ultimately occurred on a clinical basis at another site. Written consent was obtained for phenotyping and whole genome sequencing through Protocol #100 at the Utah Foundation for Biomedical Research, approved by the Independent Investigational Review Board, Inc. Informed and written consent was also obtained using the Illumina Clinical Genome Sequencing test consent form, which is a clinical test ordered by the treating physician, GJL.

The following information was supplied regarding the deposition of DNA sequences:

We have submitted the whole genome sequencing data to the Sequence Read Archive, accession number: SRP030462.

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
