# Peer review of "Integrating precision medicine in the study and clinical treatment of a severely mentally ill person"

_PeerJ, doi:10.7717/peerj.177_

## Round 0.1 · original submission · Major Revisions

As the comments from both reviewers indicate, they see a significant conceptual problem with your paper. You indicate that you see this case as constituting "an advance in the field of applying genomics to the individualized care of one person with severe mental illness." However, as the reviewers suggest--and I agree--it's not clear how the genomic findings contributed in any way to this patient's treatment. If this paper is ultimately to be acceptable to PeerJ, you will need to demonstrate that contribution clearly. Otherwise, as Reviewer 1 indicates, the genomic findings appear simply to be epiphenomenal data in a case report on DBS.

Reviewer 1 ·

Basic reporting

The report is much narrated based on the experience of one of the authors, rather than on explaining objectively the project, the work performed, and the results obtained. The discussion is subjective and repetitive in different parts and although some issues regarding availability of genomic testing and return of results are raised, these should be discussed in a more objective way.

Experimental design

No comments on the methods as most of the data production was done in an external certified clinical laboratory, data analysis was performed using commercially available software only. Maybe, more details on how the deep brain stimulation device operates and the procedures needed to implement this kind of treatment should be included.

Validity of the findings

The main and most interesting findings of this report are the favorable results obtained from the deep brain stimulation device used in the patient and the reportedly dramatic changes and improvements in the patient’s behavior. The authors would better focus on reporting, documenting, and following up in more detail the effects, both favorable and potential unfavorable, of the device in the patient’s behavior and how can this type of devices be used to treat other neuropsychiatric disorders.
This is not a genomics or whole-genome article, as the results obtained from the genomic sequencing of the patient did not informed, guided or helped in any significant way in the diagnosis or treatment of the patient. The authors do not provide any additional evidence that any of the variants associated to behavioral disorders has a contribution to the patient’s phenotype, except for the already published associations of these variants. Many “normal” individuals in the general population carry these variants and they do not present with clinical neuropsychiatric disorders, therefore the relative contribution of these variants to this patient’s phenotype is not attainable. The authors do not provide any further information, rather than speculation, on the Refsum disease variant that would support or discard the contribution of this variant to the ophthalmologic phenotype in the patient. All other genomic information from the whole-genome approach seems to be not significant for the purposes of this report and therefore treated as supplementary data. The whole-genome analysis seems to have been done superficially and although some findings can be useful for the patient’s health care in the future, such as pharmacological variants, the results obtained are not significant for this report.

Reviewer 2 ·

Basic reporting

The manuscript describes one of the first studies attempted involving whole genome sequencing as a part of “individualized medicine”. This aspect will surely attract many readers, but I believe that the manuscript in its current form would leave readers disappointed. The authors did not conclude about the usability of whole genome sequencing for diagnosis or for choosing a treatment strategy, nor did they provide personal impressions, advantages, shortcomings or prognoses of clinical use of WGS. The list of polymorphisms selected by the authors for discussion does not explain the phenotype and does not lead to any specific decisions in treatment. A possible exception is the homozygous mutation in CYP2C9, which may explain the lack of response of the patient to fluoxetine, but it needs more detailed analysis or functional assay to confirm that resistance to the drug is indeed caused solely by this variant of CYP2C9. However, even if conclusive, it does not justify WGS by itself. The conclusion section, presented in the abstract, does not contain any specific conclusion.
The introduction is written emotionally and criticizes current approaches to diagnosis (“coarse categorizations in psychiatry which bear little resemblance to reality”) and monetary incentives in clinical trials contrasting them to individualized medicine. However, it does not provide a fair introduction to the following sections.

Experimental design

Research question is not clearly defined.

Validity of the findings

Neither the original question nor the conclusions are appropriately stated.
Overall, the methods used in the study are reasonable. However, I think that the description of methods for detection of deleterious mutations creates a false feeling of reliability of the distinction between harmful and benign mutations. So far, different approaches result in lists of predicted deleterious mutations with a relatively low degree of overlap. A statement that for an Omicia Score of 0.85 the analyses results in a 1% false positive rate could be misleading without knowing the false negative rate and without specifying how these rates were calculated.

Additional comments

The usability of WGS data for clinical psychiatry at the current state of knowledge is not clear, and it is assumed that the authors will discuss it. The idea of looking for deleterious mutations is reasonable, but it was not justified or explained in the manuscript. We know that OCD is not a monogenic disorder, so identification of one deleterious mutation even in a very good candidate gene will not “explain” the disease. So what was the purpose of WGS, what were the expectations, and what models could be used for the analysis? I think that the manuscript should be revised. I recommend clearly stating the aim of the study and conclusions. WGS data needs a different angle of discussion, specifically how sequencing can help in diagnosis and selection of an optimal treatment strategy.

---

## Round 0.2 · Major Revisions

We have now received a re-review of your paper from one of the original reviewers. As you can see from the comments, the reviewer has indicated that further major revision is required, and I agree. The problem that remains with this paper is the disjunction between your assertions that the case study demonstrates the utility of integrating WGS into this patient’s treatment and the difficulty in identifying how the WGS findings impacted that treatment. As has been the case from the beginning, your challenge is to demonstrate what your title promises, i.e., that precision medicine in fact has been integrated into and made a difference in this patient’s treatment. If you would like to have another shot at supporting that claim, we would entertain another round of revisions. However, if you choose to seek publication elsewhere, we would certainly understand.

Reviewer 1 ·

Basic reporting

No comments

Experimental design

WGS is not clearly justified: what was the goal / research question of WGS?

Validity of the findings

• The conclusions should be appropriately stated, should be connected to the original question investigated, and should be limited to those supported by the results.

“To our knowledge, this is the first study in the clinical neurosciences that integrates detailed neuropsychiatric phenotyping, deep brain stimulation for OCD and clinical-grade WGS with management of genetic results in the medical treatment of one person with severe mental illness.” (Conclusions section of the Abstract).
It is not clear what the authors mean by “management of genetic results in the medical treatment”; there is no mention of any influence of genetic study on diagnosis, prognosis or medical treatment. And "integration" of "clinical-grade WGS" with other parts of the study is overstated; the authors barely try making reasonable connection between sequencing results and the phenotype.

“We offer this as an example of precision medicine in neuropsychiatry including brain-implantable devices and genomics-guided preventive health care.”
Genomics did not guide the authors to any specific medical decisions, and they did not describe how genomics helped them in the prevention of any health-related issues of the patient (see also my remarks about Conclusions section of the manuscript). The authors actually offer opinions within the manuscript which one could consider as not fully agreeing with the statement above: “WGS cannot act as a diagnostic and prognostic panacea, but instead could act to elucidate potential risk factors for some illnesses”. It is not clear how the authors make a clear distinction between “potential risk factors” and prognosis, and if they think knowing “potential risk factors” provides sufficient information to guide preventive health care.

First paragraph of Introduction criticizes current standardized approaches in current medicine and psychiatry in particular, and contrasts it with individual-focused medical care. It sounds like a manifesto making bold general statements, most of which could be considered controversial. It is inappropriate for a case study, and does little to introduce specific problems addressed in the manuscript.
There are no objections to the description and short discussion of genetic variants found in the patient and their possible involvement in the phenotype. Personal genome interpretation is still in its infancy, and every case is an interesting teaching point. However, more general discussion of WGS and conclusions are sometimes irrelevant to the case. The motivation of using WGS for the patient is unclear.
The examples of use of whole genome or exome sequencing for identification of disease cause (Introduction) are deceiving, as they are related to monogenic disorders when only one mutation in one gene is causative. This paradigm is not applicable to OCD. OCD is a complex disorder, with at least several genetic factors involved in the etiology. Based on genome-wide association studies, it is quite possible that some genetic variations predisposing to the disease are not missense or nonsense mutations in protein-coding genes, but are located outside known genes and involved in molecular pathology through yet unknown mechanisms.
Another misleading example is presented in the Conclusion section, where the authors refer to data of other groups using WGS as a part of the evaluation of patients with undiagnosed diseases or diseases with unclear etiology. It was expected that the disease in every particular case could be caused by either one or a combination of very few genetic variants in the genes known to be involved in diseases with similar phenotypes. The motivation of referred articles was to improve diagnosis of, or the risk to develop, a particular phenotype by identification of such specific DNA variants or mutations. It does not seem to be a goal in the described study, and again raises the question: what was the reason of WGS in this particular patient?

“…our data also does not allow us to exclude the possibility of polygenic and epistatic modes of inheritance” (397-398) – I would exclude this phrase; although the statement is formally correct, there is a notion that the authors tried or wanted to exclude polygenic mode of inheritance; I hope they did not.

“We provide our study as a cautionary one: WGS cannot act as a diagnostic and prognostic panacea, but instead could act to elucidate potential risk factors for some illnesses” (398-400). The authors discussed using WGS for different kinds of disorders, including monogenic disorders. For many of monogenic disorders WGS may provide all the necessary information for diagnosis and, in many cases, prognosis of the disease and treatment. The authors make this statement in a context which may assume only neuropsychiatric disease, but it would be more appropriate to specifically state it.

The authors provide the only example of usefulness of WGS for the patient: “health information stemming from these data were nevertheless immediately useful in the care of this person, as a variant associated with his ophthalmologic phenotype did indeed inform and enrich his care” (Conclusions, 477-479). However, in the manuscript (without consideration of supplemental material) there is no mentioning of any mutation involved in ophthalmologic phenotype, and there is no information about how these data were used in the care of the patient.

Overall, the manuscript describes an interesting study which deserves publication. However, it contains several confusing discussion points and still lacks clarity in the goal of WGS of the genome of the patient. The manuscript would benefit tremendously from shortening the introduction, discussion and conclusions to strictly relevant points, and making the manuscript more focused and concise.

---

## Round 0.3 · accepted · Accept

With this new version of the paper, I think you have successfully addressed the reviewer's comments. In particular, as you now make clear, this case illuminates the complexity of integrating WGS data into clinical care.